# The Effectiveness of an Adaptive Method to Analyse the Transition between Tumour and Peritumour for Answering Two Clinical Questions in Cancer Imaging

**DOI:** 10.3390/s24041156

**Published:** 2024-02-09

**Authors:** Margherita Mottola, Rita Golfieri, Alessandro Bevilacqua

**Affiliations:** 1Alma Mater Research Institute on Global Challenges and Climate Change (Alma Climate), University of Bologna, 40126 Bologna, Italy; margherita.mottola@unibo.it; 2Department of Medical and Surgical Sciences (DIMEC), University of Bologna, 40138 Bologna, Italy; rita.golfieri@unibo.it; 3Department of Computer Science and Engineering (DISI), University of Bologna, 40126 Bologna, Italy; 4Advanced Research Center on Electronic Systems (ARCES), University of Bologna, 40125 Bologna, Italy

**Keywords:** machine learning, peritumour, tumour, hepatocellular carcinoma, rectal cancer

## Abstract

Based on the well-known role of peritumour characterization in cancer imaging to improve the early diagnosis and timeliness of clinical decisions, this study innovated a state-of-the-art approach for peritumour analysis, mainly relying on extending tumour segmentation by a predefined fixed size. We present a novel, adaptive method to investigate the zone of transition, bestriding tumour and peritumour, thought of as an annular-like shaped area, and detected by analysing gradient variations along tumour edges. For method validation, we applied it on two datasets (hepatocellular carcinoma and locally advanced rectal cancer) imaged by different modalities and exploited the zone of transition regions as well as the peritumour ones derived by adopting the literature approach for building predictive models. To measure the zone of transition’s benefits, we compared the predictivity of models relying on both “standard” and novel peritumour regions. The main comparison metrics were informedness, specificity and sensitivity. As regards hepatocellular carcinoma, having circular and regular shape, all models showed similar performance (informedness = 0.69, sensitivity = 84%, specificity = 85%). As regards locally advanced rectal cancer, with jagged contours, the zone of transition led to the best informedness of 0.68 (sensitivity = 89%, specificity = 79%). The zone of transition advantages include detecting the peritumour adaptively, even when not visually noticeable, and minimizing the risk (higher in the literature approach) of including adjacent diverse structures, which was clearly highlighted during image gradient analysis.

## 1. Introduction

Recent advances in computer science and engineering have increased the availability of high-performance computing resources and have favoured a widespread use of machine learning (ML) techniques in medical image analysis to complement the traditional visual-based assessment of images with quantitative measurements of image latent properties, the so-called radiomics features [1]. In this approach, imaging features are exploited by classification models for the early prediction of disease, prognosis, and response to therapy [2], with the aim of improving the accuracy of clinical decisions and facilitating timely clinical actions [3]. In particular, many applications of ML techniques refer to oncology, aiming at disclosing tissue properties to support precision medicine in the fight against cancer [4]. In this regard, although quantitative radiomics analysis is primarily focused on the visually detected tumour boundary, several recent works have dealt also with the peritumour area, the site where the tissue changes start earlier, and whose role is known to be crucial in determining the tumour behaviour in terms of progression and response to therapy [5]. In fact, the extent of the peritumour region, which may have an influence on diversity of tumour aspects, is not known a priori. In addition, the peritumour is often not visually detectable by radiologists because the inflammatory tissue, characterizing the transition between tumour and normal parenchyma, makes the radiological imaging findings misleading [5]. The works in the literature propose an extensive array of solutions for peritumour detection and analysis, relying on a priori-defined analytical choices [6] and sometimes even on manual intervention [7], but never exploiting adaptive strategies. However, while these approaches can work with single, particular cases of application, they are very far from representing a methodological approach. In fact, the complexity of clinical issues cannot be faced with end-to-end solutions but instead require robust methodologies to be parametrized for diverse tumour sites and imaging modalities.

In this study, we developed an adaptive method to automatically detect the transition region bestriding tumour and peritumour, exploiting local image contrast variations that were tested on two different tumour types, the hepatocellular carcinoma (HCC), imaged by contrast-enhanced computed tomography (CT), and locally advanced rectal cancer (LARC), investigated through magnetic resonance imaging (MRI). In the former case study, we addressed the early diagnosis of HCC nodules with microvascular invasion (MVI), a consolidated predictor of HCC recurrence after curative treatments [8]; in the latter, the early prediction of LARCs responding to neoadjuvant chemo-radiotherapy (nCRT) was adopted [9]. To this end, we developed ML models based on explainable imaging features, and the results were compared with those achieved by exploiting a common representative nonadaptive approach [10].

The paper is organized as follows. Section 2 discusses the state-of-the-art the solutions proposed for the analysis of peritumour regions. Section 3 presents the method we conceived, the dataset used, and the methodology adopted to develop the predictive models. Section 4 reports the experimental results. Further discussion is provided in Section 5, while Section 6 draws conclusions and provides some indications for future work.

## 2. Literature Review

A great number of works exist on integrating the quantitative analysis of the tumour core with that of the surrounding regions for many solid tumour type (e.g., lung [11,12,13,14,15,16], liver [10,17,18,19,20], rectum [7], tongue [21], brain [22,23], breast [6,24,25], cervix [26], oesophagus [27]). In addition, almost all of these address two imaging modalities only, CT [6] and MRI [26], which can offer the highest imaging resolutions. The bulk of these studies have analysed the peritumour area by extending the original tumour segmentation by variable and prefixed extents and excluding the tumour core. Of course, this approach relies on a clear separation between the tumour and peritumour, which is not feasible from a clinical point of view because the inflammation around the tumour is responsible for a stepwise transition between tumour and normal tissue. In addition, the peritumour segmentations achieved through single [13,14,17,25,28] or multiple measures [6,12,15,16,21,26] also fail across multiple tumour types. such as for lung peritumour parenchyma, analysed from 2.5 mm [16] to 20 mm [15] from the tumour margins. This is also relevant to HCC peritumour analysis aimed at MVI prediction using MRI imaging [18], performed at 10 mm [18] or even 20 mm [17] of the radius from tumour borders. As a consequence, such a variability prevents any methodological comparison between these works. Some other studies have even exploited the manual segmentation of the peritumour [7,20,22,23,27] in cases where the inflammatory tissue is (apparently) easily distinguishable from the normal one through the grey level (GL) scale. Nevertheless, this approach is prone to all weaknesses of manual segmentation, remaining limited to the specific application and lacking in any generalizability. There are a few studies that have focused on the lung [11] and liver [10,19] that have addressed the transition zone between the tumour and peritumour regions [10,11,19] where changes have occurred and thus have more clinical relevance than do analyses derived from the peritumour alone. What these works have in common is a strategy for analysing the so-called tumour rim (hereinafter, tRIM), stemming from the morphological segmentation of the original tumour ROIs, achieved by subtracting its erosion from dilation, both achieved with a variable size of the structuring element (SE), such as 2 mm [10,19] or 3 mm [11]. In fact, the tRIM arises from binary masks of tumour ROIs, independently from any GL value in the tumour and surrounding tissue. Besides being prone to including adjacent structures of a different nature, it is unlikely that the underlying region could have a strong relation with the physical transition zone, where GL shades are of utmost importance for its detection.

## 3. Materials and Methods

### 3.1. Study Population

This study is based on two different, single-centre, retrospective datasets acquired from the Radiology Unit of the IRCCS Polyclinic Sant’Orsola-Malpighi, University of Bologna, Bologna, Italy.

The first dataset includes 117 patients with HCC imaged by contrast-enhanced CT, who underwent surgery [29] (hereinafter, HCC-CT dataset). The enrolled patients fulfilled the following criteria: (a) preoperative CT performed in our Radiology Unit within three months before surgical resection, (b) HCC imaging diagnosis reached according to EASL guidelines [30], (c) nodule dimension ≤ 3 cm, and (d) hepatic resection indicated according to the criteria described in [31]. In summary, 12 patients who received previous treatments, 24 who underwent preoperative imaging performed outside our radiology unit, and 3 who underwent inadequate imaging studies were excluded. Finally, the analysis was performed on 78 patients comprising 89 distinct HCC nodules, 32 of which were diagnosed with a positive MVI status (MVI+) and 57 with a negative MVI status (MVI−). The protocol’s requirements for contrast-enhanced CT met the criteria recommended by the EASL guidelines [30]. More details on patients’ characteristics and the technical specifications of the CT examinations can be found in [32].

The second dataset consists of 91 patients with LARC who underwent MRI for primary staging (hereinafter, LARC-MRI dataset). The enrolled patients met the following criteria: (a) diagnosis of LARC at our institution performed through (b) pretreatment MRI for primary staging and (c) treatment with long-course nCRT followed by (d) total mesorectum excision (TME). At the time of TME, the histopathological reports provided the tumour regression grade (TRG) according to the TRG staging system of the American Joint Committee on Cancer (AJCC) [33]. Then, 7 patients without available TRG information in the pathological report, 11 patients who did not undergo CRT, 24 patients with surgical resection performed outside our institution, and 3 patients who underwent MRI with incomplete staging or imaging artefacts were excluded. Finally, the analysis was carried out on 46 patients, 18 of whom had TRG 0 and TRG 1 (TRG [0–1])—namely responders (R)—and 28 with TRG 2 and TRG 3 (TRG [2–3]), namely nonresponders (NR). More details on patients’ characteristics and technical specifications of MRI examinations can be found in [9].

### 3.2. Study Workflow

Figure 1 shows the block diagram summarizing the workflow adopted for this study, starting from the HCC and LARC datasets (Figure 1a). The workflow is composed of four macro steps, each described in the following sections and included tumour segmentation (Figure 1b, Section 3.3), ZoT and tRIM detections (Figure 1c, Section 3.4 and Section 3.5), feature generation (Figure 1d, Section 3.6.1), and HCC and LARC studies (Figure 1e), Section 3.6.2, which were Section 3.6.3 carried out by exploiting features generated from ZoT and tRIM, respectively.

### 3.3. Tumour Segmentation

Manual segmentation of tumour ROIs was performed by two experienced radiologists in consensus [34]. Regarding the HCC-CT dataset, for each patient slice, HCC ROIs were drawn on both arterial- and delayed-phase CT images, along the visible lesion’s boundaries, in order to include approximately the entire lesion volume. More details can be found in our previous work [29]. As regards the LARC-MRI dataset, tumour ROIs were outlined slice by slice on a T2-weighted (T2w) series on the 3 mm-thickness axial plane, with the intestinal lumen being excluded [9]. For both datasets, tumour segmentation was performed using ImageJ software (https://imagej.nih.gov/ij/, R1.53d, accessed on 20 March 2022), a freeware and open-source software developed by the National Institutes of Health (NIH). When performing tumour segmentation, radiologists were always blinded to the histopathological reports.

### 3.4. A Method to Detect the Zone of Transition

The ZoT [35] between the tumour core and peritumour area can be thought as an annular-like-shaped area surrounding tumour, where tissue characteristics maintain the highest uncertainty. We previously exploited the concept of ZoT in a clinical work [29], employing an exploratory method and highlighting the potential benefits of the ZoT analysis in the clinical management of early-stage HCC. The goal of this work was to provide a detailed methodology to define the ZoT extent after detecting the ZoT’s outer and inner borders. To this aim, we built a pipeline to deeply analyse the magnitude of variations along the directions of the tumour border’s gradient.

Figure 2 depicts the four main steps of the procedure, whose inputs are the ROIs segmented on the GL slices for (Figure 2a) LARC and (Figure 2b) HCC (during the arterial phase). The first step consists of the morphological edge detection performed on the tumour ROI through a square 3×3 SE for (c) LARC and (d) HCC. The second step is the gradient profile analysis that is performed on the gradient magnitude (GM) image achieved through a 3×3 Sobel kernel (e,f)). In practice, (g) a thin, 2-pixel width, 15 mm-long stripe is centred perpendicularly on each pixel of the tumour border and at $\pm 45^{\circ }$ to cope with the digital representation of the border. In particular, 15 mm is a heuristic parameter determined after evaluating tumour ROIs to the maximum extent expected for contrast variations alongside the ROI edge. Hence, 15 mm was chosen as being sufficiently long to include the widest contrast variations along the tumour border and as being sufficiently short to reduce the computational burden. The example in Figure 2e,f was performed with 15 × 2-pixel stripe (i.e., by hypothesizing that 1 pixel measures 1 mm), where the small coloured dots represent the set of displacements exploited in our analysis. For denoising purposes, each couple is averaged to finally achieve a 1-pixel width stripe (h), whose gradient profile is depicted in Figure 2i. As expected, the gradient profile is bell-shaped, with its maximum value at the edge pixel (the cyan dot in (i)). Indeed, on the left and the right sides, they show either a sudden change in slope (e.g., “a corner”, the red dot) or even a local minimum (the orange dot), depending on whether the outer and inner tumour regions are (differently) homogeneous or show structural variations, respectively. These two points, the orange and the red ones, represent the extent of the *a gradient transition line*; that is, a line-wise ZoT to be detected for each stripe displacement. To improve the accuracy of minima or corner detection, the gradient profiles are interpolated using a nonparametric smoothing spline [36], with a step of 0.1. Accordingly, the GM images are interpolated using the Lanczos interpolation method [37], with the same step size. Although localizing the minima is trivial, to detect the red points, we exploit the pruning algorithm proposed by [38] to find the corner point of the *L*-curve (as one can see, the rapidly changing slope recalls the *L*-curve trend). Finally, the detected transition line is reported as a binary mask (Figure 2j). Sometimes the maximum value of the transition does not occur in correspondence to the tumour border (i.e., the central pixel of the transition line). Accordingly, these possible small displacements between the maximum GM and edge’s pixels are compensated for by shifting, thus keeping as the reference border the radiologist’s tumour segmentation.

After the stripes are processed, the outcome is a sparse ZoT in the terms of a binary mask. The third step of the pipeline is the mask thickening, realized through a morphological dilation with a 9-pixel-width square SE, whose result is depicted in Figure 2k for LARC and (l) for HCC.

Finally, the binary masks are fed to the fourth stage to compute the density maps [39], as shown in the pseudo-colours in Figure 2m,n. The automatic segmentation of the density maps, performed by exploiting the active contouring of Chan–Vese algorithm [40], allows for the defining of the black borders shown in the same figure, where they have been linearly rescaled at the original resolution. This concludes the procedure and produces the definite ZoT.

The entire pipeline was implemented in the Matlab^®^ environment (R2019b v.9.7, The MathWorks, Natick, MA, USA).

### 3.5. tRIM: The Alternative to ZoT

The most popular approach to semi-automatically select the zone of transition between tumour and peritumour is represented by tRIM. Usually, it considers rings of a different extent, overlapping the tumour margin, generated via common morphological operations starting from the tumour ROI. In particular, these rings have a different inner and outer radial distance from the tumour border. In all studies, the lengths of these radii is always defined a priori, trying single or multiple values, with not any semantic reference with the underlying tissue structures. To perform a fair comparison between ZoT and tRIM, we need to have the two annular regions as much similar as possible. Since our method defines the ZoT ROI adaptively, only tRIM ROI can be changed. Accordingly, we have conducted a set of preliminary tests with different pairs of inner and outer radii for tRIM, for each slice, comparing each corresponding annular region with ZoT’s. At the end, we have chosen the pair for which the two regions most frequently match, that is 6 mm and 3 mm as outer and inner radius for HCC-CT, and 3 mm for both radii for LARC-MRI.

### 3.6. ML Models

To assess the benefits of using ZoT instead of tRIM in answering the two clinical questions considered via the datasets, we have a pool of imaging features from ZoT and tRIM as well as from the tumour core and investigated their relevance in the early prediction of a MVI+ status for HCC and for the treatment efficacy in LARC. To this purpose, we carried out different studies, involving many ML models aimed at assessing, separately, the contributions of the ROIs’ shape; that is, ZoT, tRIM, and tumour. Furthermore, since ZoT ROIs are derived from GM, we thought that GM images could also yield useful information and, as such, they were also used to generate “explainable” features referring to ZoT and tRIM. To favour the interpretability of results, the developed ML models used features in a low dimensional space.

For the HCC-CT dataset, we conducted four different studies (hereinafter, ${\mathrm{HCC}}_{\left[1–4\right]}$) investigating, both the arterial and venous CT phases and the role of the transition zone using tRIM (${\mathrm{HCC}}_{\left[1,2\right]}$) or ZoT (${\mathrm{HCC}}_{\left[3,4\right]}$) ROIs, which were consistently combined with the analysis of tumour ROIs. In particular, the analysis was performed on GM images in ${\mathrm{HCC}}_{\left[1,3\right]}$ studies and on the GL series in ${\mathrm{HCC}}_{\left[2,3\right]}$. As regards the LARC-MRI dataset, its limited size allows for the exploring of the discriminative power of the single features only, rather than their combinations. Therefore, we carried out five studies (hereinafter, ${\mathrm{LARC}}_{\left[1–5\right]}$), referring to either tRIM (${\mathrm{LARC}}_{\left[1,2\right]}$), ZoT (LARC${\mathrm{LARC}}_{\left[3,4\right]}$), or tumour (${\mathrm{LARC}}_{5}$) ROIs. Similarly to the HCC studies, we analysed GM in ${\mathrm{LARC}}_{\left[1,3\right]}$ and the GL images in ${\mathrm{LARC}}_{\left[2,4,5\right]}$.

For all image processing procedures, the implementation environment for ML models and statistical analyses was Matlab^®^.

#### 3.6.1. Feature Generation

Twelve statistical descriptors were considered for first-order feature generation: Shannon’s entropy (E), mean (μ), standard deviation (σ), median (M), median absolute deviation (MAD), skewness (S), kurtosis (K), uniformity (U), interquartile range (IQR), coefficient of variation (CV), mean (μ90th), and the median (M90th) of the last decile. The first ten were computed slice by slice to achieve as many parametric maps as possible via the local approach proposed in [41]. Subsequently, all twelve were used for generating single-value features from these maps, thus producing 120 features in total. When tumour ROIs were used, all these features arose from the original GL images only. As mentioned above, when ZoT and tRIM were used, the ten local maps were also computed on the GM series, yielding 240 features each for ZoT and tRIM. In addition, for both ZoT and tRIM, the twelve descriptors were also computed on both the original GL and GM series, thus yielding 132 features each. Figure 3 summarizes the features considered for ${\mathrm{HCC}}_{\left[1-4\right]}$ and ${\mathrm{LARC}}_{\left[1-5\right]}$ studies.

As regards ${\mathrm{HCC}}_{\left[1–4\right]}$, each study referred to a pool of 762 imaging features; these were (i) 264 arising from tRIM or ZoT (i.e., 132 features each) for the arterial and venous phases; (ii) 240 stemming from tumour ROIs (i.e., 120 features for each CT phase); (iii) 6 features, measuring the volume (mm^3^) of the ROIs of the ZoT, tRIM, and tumour in the arterial and venous phases, respectively (i.e., 4 features), along with their ratio (i.e., 2 features); and (iv) 252 features originating from the comparison between the arterial and venous phases (Figure 3). Specifically, these latter 252 features referred to the signed relative change (RCS) between the two phases and has been formulated as in Equations (Equation 1)–(Equation 3):(1)$$RCS=\frac{|f_{A}-f_{V}|}{\bar{f}}\cdot \mathrm{sgn}\;f$$
(2)$$\bar{f}=\left|mean\left(f_{A},f_{V}\right)\right|$$
(3)$$\mathrm{sgn}\;f=\left\{\begin{array}{cc} +1 & \mathrm{if}\;(f_{A}-f_{V})\geq 0\\ -1 & \mathrm{if}\;(f_{A}-f_{V})<0 \end{array}\right.$$
where *f* is a generic feature; $f_{A}$ and $f_{V}$ are the values of each single feature on arterial and venous phases, respectively; $\bar{f}$ is the absolute mean between $f_{A}$ and $f_{V}$ (Equation (Equation 2)); and sgn *f* is the *sign* function.

Finally, in the LARC studies, a set of 132 features were considered for ${\mathrm{LARC}}_{\left[1–4\right]}$, originating from the tRIM and ZoT analysis performed on the GM or GL series, respectively, as were a set of 120 features for ${\mathrm{LARC}}_{5}$, referring to tumour analysis (Figure 3).

#### 3.6.2. HCC-CT Studies

#### Feature Selection

After preliminary discriminative analyses with 3-dim features and after considering the size of the original HCC-CT dataset, we decided to exploit a 4-dim space. The feature selection step was performed after data normalization through the linear scaling of each single feature between the first and the third quartiles of the distribution and data standardization. First, a subset of the most relevant imaging features were preliminary selected through the least absolute shrinkage and selection operator (LASSO) method, which exploited 5-fold cross-validation (CV) on the entire dataset and weighed each sample by its prior probability. To derive the most discriminative subset made of k=4 features (i.e., a quadruple) only, all possible quadruples given by the Binomial coefficient nk, where *n* is the number of features initially returned by LASSO, were considered. Then, those quadruples where at least one couple had a linear correlation coefficient ρ ≥ 0.3 were discarded, and the remaining ones were used by a support vector machine (SVM) to determine the different separating hyperplanes. The discrimination achieved by each hyperplane was tested for statistical significance (α = $10^{-3}$) through the Wilcoxon rank-sum test with Holm–Bonferroni correction. Finally, of the resulting significant quadruples, the one yielding the highest informedness (I) of the receiver operating characteristic (ROC) curve was selected.

#### SVM Classifiers

First, the quadruple selected for each study (${\mathrm{HCC}}_{\left[1–4\right]}$) was augmented to increase the statistical significance of the training and validation subsets used to set up the SVM classifier. To this aim, the univariate kernel density estimation (KDE) of each single feature was considered, and the Latin hypercube sampling (LHS) method [42] was used for data oversampling through $10^{5}$ runs, with the correlation of oversampled variables being controlled for. Then, the optimal LHS solution was searched to minimize the correlation difference between the original and oversampled variables through a properly developed cost function (more details in [29]). Finally, the oversampled HCC-CT dataset consisted of 169 samples—62 MVI+ and 107 MVI−—thus preserving the proportion of positive and negative instances in the original dataset. Table 1 is the dataset description and includes the original and oversampled datasets.

One SVM classifier was developed for each study, where MVI+ and MVI− instances were considered as true positive (TP) and true negative (TN), respectively. The entire dataset was partitioned into a training and holdout test set. The training set of 117 samples (43 MVI+ and 74 MVI−) was extracted according to the method proposed by [43], which we used in [41], which allows for the preserving of the representativeness of the entire dataset within both training and test sets. According to this method, an SVM is initially applied on the entire dataset to find the support vectors, which are split into three classes based on their SVM margin (${\mathrm{SVM}}_{m}$). That is, for each sample, ${\mathrm{SVM}}_{m}$> 1 if correctly classified, and ${\mathrm{SVM}}_{m}$< 0 if misclassified, while 0 ≤ ${\mathrm{SVM}}_{m}$≤ 1 if it falls within the decision surface. Subsequently, the samples were randomly assigned to the training set by preserving the proportion of each class within the entire dataset. Accordingly, the test set, consisting of the leftover 52 samples, had the same proportion. SVM hyperparameter tuning was carried out by exploiting 100 runs of 3 fold-CV, where each fold had 39 samples, 14 MVI+ and 25 MVI−, randomly assigned. Because of the class unbalancing, the prior probability of MVI+ and MVI− samples was used to weigh the SVM misclassification cost. For each run, the SVM hyperparameters, linear scale γ, and misclassification cost C were estimated using the built-in MatLab Bayesian optimization algorithm [44], and the predicted probability and label of each sample was estimated using a binomial logit function. The ROC curve was built, and its area under the curve (AUC) was used to assess the performance of each CV run. Hence, from each run, the model selection was performed in two steps: (i) the models having an AUC in training folds lower than that in the test folds were discarded since most are prone to overfitting, and (ii) the models with the highest AUC on the training folds were kept. Then, the surviving models (100 at most) were trained on the three folds together, and the ultimate model was selected as the one yielding the highest I and AUC. Finally, it was externally validated on the holdout set.

#### Comparison among Studies

The ROC curves of the four SVM classifiers arising from the studies ${\mathrm{HCC}}_{\left[1–4\right]}$ were compared to determine whether (i) the features generated from ZoT improve the prediction of MVI+ HCC nodules with respect to the standard tRIM and (ii) whether the analysis of GM images is more informative than is the analysis of the original GL series. First, the statistical significance of each classifier was tested by Wilcoxon rank-sum test at $\alpha =10^{-3}$ and represented through box plots. In addition, different metrics were considered, including AUC, the number of false-positive (FP) errors, the number of false-negative (FN) errors, accuracy (ACC), and especially I, which provides the balance at the Youden cutoff between sensitivity (SN) and specificity (SP). To assess the clinical utility of the developed SVM classifiers, the negative and positive predictive values (NPV and PPV, respectively), and the diagnostic odds ratio (DOR) were also measured.

#### 3.6.3. LARC-MRI Studies

A univariate statistical analysis was performed on each of the five studies (${\mathrm{LARC}}_{\left[1–5\right]}$) to identify the most discriminant feature in predicting R patients (considered as TP) through the Wilcoxon rank-sum test, with α = 0.05 as the level of significance being used along with Holm–Bonferroni correction. Median and IQR values of the features selected in each study were compared between the two groups of R and NR patients and are also represented through box plots. The ROC curves of all single features and their AUCs are reported. In particular, the I, SN, and SP values were compared. For the sake of clarity, Table 2 summarise the dataset description.

## 4. Results

### 4.1. Visual Comparison between the ZoT and tRIM ROIs

Figure 4 reports a panel of tRIM and ZoT ROIs for a few representative slices of the HCC-CT and LARC-MRI datasets.

The tRIM area is located between the two green lines, whilst the ZoT is the area located between the two red lines. As for the tumour core, it is bounded by the blue line and highlighted by a shade of blue. From a to d, Figure 4 shows a few slices of MVI+ (a,b) and MVI− (c,d) nodules both in the arterial (b,c) and venous (a,d) phases. Similarly, Figure 4e–l shows a few LARC slices, both in cases where ROIs have quite regular shapes and large sizes (e–g) and in cases where ROIs have irregular borders (h–l). In addition, the pink arrows in Figure 4h–l point out the key differences between the tRIM and ZoT ROIs, highlighting the most irregular parts of tumour edges.

### 4.2. Diagnosis of MVI in HCC

Table 3 reports the results of the feature selection procedure performed on all studies ${\mathrm{HCC}}_{\left[1–4\right]}$, while Table 4 indicates, for each identification code in Table 3, the corresponding feature’s name.

Regarding the tRIM studies (${\mathrm{HCC}}_{\left[1,2\right]}$), the feature selection stage yielded 26 and 21 significant quadruples, respectively. In contrast, as regards the studies involving ZoT, 18 and 40 quadruples remained, respectively. The most discriminant quadruples for each study had the same *p* ∼ $10^{-9}$. Each quadruple in Table 3 has the performance reported in Table 5 and Table 6 in training (117 samples) and testing (52 samples).

In addition, Figure 5 reports the ROC curves in the training (Figure 5a) and test (Figure 5b) sets, with the Youden cutoff highlighted by black circles. Finally, Figure 6 and Figure 7 show the box plots of all models in both the training and test sets, where all models have at least *p*∼$10^{-12}$ and *p*∼$10^{-5}$ in the training and test sets, respectively. Box plots report nonoverlapping interquartile ranges, with the median values of single groups differing more than the 60% between MVI+ and MVI−.

### 4.3. Detection of TRG [0–1] in 

From the three ${\mathrm{LARC}}_{\left[1,3,5\right]}$ studies, three single features were selected with *p* < $10^{-4}$, and their box plots are reported in Figure 8; meanwhile, no significant features resulted from the tRIM and ZoT analyses of the GL images (i.e., ${\mathrm{LARC}}_{\left[2,4\right]}$).

Tumour analysis was performed via S−E (Figure 8a), which separated the R and NR groups with *p*∼10$10^{-5}$, while the {median, IQR} values were equal to {0.90, 0.80} in R and to {−0.36, 1.26} in NR. Peritumour analysis was performed through σ−S (Figure 8b) from tRIM and σ−K (Figure 8c) from ZoT, which both separated the R and NR groups with *p*∼$10^{-4}$. Then, σ−S had a {median, IQR} equal to {−0.54, 1.00} in R and to {0.22, 0.89} in NR, whilst σ−K had a {median, IQR} equal to {−0.56, 0.39} in R and to {−0.05, 0.76} in NR.

Table 7 reports the discriminative performance between the R and NR groups of S−E, σ−S, and σ−K, referring to the ${\mathrm{LARC}}_{\left[1,3,5\right]}$ studies, respectively; meanwhile, Figure 9 reports the corresponding ROC curves with AUC = 0.86 (95% CI, 0.63–0.92) for S−E, AUC = 0.81 (95% CI, 0.65–0.91) for σ−S, and AUC = 0.82 (95% CI, 0.63–0.93) for σ−K.

From tumour analysis, S−E yielded an I = 0.62, corresponding to SN = 72% and SP = 89%, while σ−S, stemming from the tRIM investigation, had an I = 0.58, which corresponding to SN = 83% and SP = 75%. In contrast, σ−K, originating from the ZoT, differentiated the two groups with I = 0.68, corresponding tp SN = 89% and SP = 79%.

## 5. Discussion

The analysis of peritumour has been acknowledged as being of a wide importance to obtaining useful information regarding the transition between inflammatory and cancer tissue. So far, the detection of the peritumour area has relied upon two main alternative approaches: (i) when applicable, the manual segmentation of the peritumour based on tumour appearance; and (ii) the dilation of the binary tumour segmentation mask using a prefixed size, disregarding the information retained by the underlying GL images. In this study, we present the first adaptive method based on image contrast variations to automatically detect the ZoT boundaries, conceived to work with different tumour tissues and scanning technologies. The performance of the method was assessed on a CT and an MRI dataset, the HCC and LARC, respectively, showing a markedly different radiological appearance. Indeed, while HCCs are nearly circular small nodules with shaded borders, LARCs have a larger size, with a well-defined anisotropic, jagged shape. Hence, in cases where nodules have a regular, nearly circular shape (Figure 4a–g), the ZoT adaptive detection appeared highly similar to that of tRIM, with intersecting edges often overlapping. Nevertheless, despite their similarity, being anisotropic, the ZoT ROIs included uneven portions of the peritumour area. This led to some ZoT margins overlapping the tumour ones. For instance, in Figure 4e, the ZoT includes more peritumour in the lower and the right sides than in the upper and the left ones, which was caused by a variability in the local image contrast. The same phenomenon appears in Figure 4f,g. On the other hand, in the presence of small concavities (Figure 4h–l), tRIM shows its weakness, since its underlying morphological operations cannot follow the anatomical profiles, as emphasized by the straight lines, and, even worse, it may include adjacent structures of a different nature that could yield misleading features.

The ML model was enriched with imaging features generated from the ZoT and the tRIM ROIs, respectively, and the outcome was compared based on standard clinical predictive metrics. In the experiments, all ML models automatically selected features from the ZoT and tRIM, thus confirming the active contribution of the peritumour inclusion in building the predictive models.

In particular, as regards the diagnosis of MVI in HCC, all selected quadruples referred to the arterial and venous CT phases, respectively and jointly by means of RCS measurements. The best discrimination between MVI+ and MVI− of the entire dataset was achieved equally by the ${\mathrm{HCC}}_{\left[2,3\right]}$ studies (with I = 0.74). Both the quadruples of the ${\mathrm{HCC}}_{\left[2,3\right]}$ studies contain F136 and F754, originating, respectively, from the tumour core in the arterial phase or from the RCS measurement. In addition, one more feature (F466 in ${\mathrm{HCC}}_{2}$, F465 in ${\mathrm{HCC}}_{3}$) refers to the venous phase, while the last feature arises from the analysis of the transition using the tRIM (F592 in ${\mathrm{HCC}}_{2}$) or ZoT (F72 in ${\mathrm{HCC}}_{3}$) ROIs. Globally, the SVM classifiers of all HCC studies exhibited similar values for all ROC-related metrics as well as comparable trends of the ROC curves (with the highest I being equal to 0.74 in the training for ${\mathrm{HCC}}_{3}$ and 0.69 in the testing for ${\mathrm{HCC}}_{\left[1–3\right]}$). A partial exception is represented by ${\mathrm{HCC}}_{4}$, which had a slightly inferior performance in the test set. In fact, HCC nodules, being circular, weaken the benefits of an adaptive ZoT shape, and the SVM classifiers show a similar behaviour. It is worth noting that in both the training and validation, all models showed excellent values of NPV. which is, among the metrics used to assess model performance in clinical settings, the most relevant one for the MVI prediction because it allows for selecting the patients that mainly can benefit from surgical treatments, with a very high probability of success. The last remarks relate to the visual assessment of the box plots of all models in both the training and test sets (Figure 6 and Figure 7), which demonstrate the excellent separation between the MVI+ and MVI− groups.

As regards the detection of the R group in LARC studies, tumour analysis showed–as expected—the NR group as being more heterogeneous than the R one. This was also confirmed by the peritumour analysis performed via ZoT, which also yielded lower IQR values for both groups. It is worth noting that ZoT analysis reverses SN and SP values with respect to those achieved through tumour analysis. On the one hand, tumour core analysis yielding a higher SP than SN can be reasonably more effective in detecting NRs, as they are supposed to be at a late stage of progression [9], thus making their response to therapy challenging. On the other hand, ZoT analysis, which provided a higher SN than SP, may perform better in detecting Rs, considered to be at an early disease stage [9], thus boosting its potential role for identifying the best candidates to nCRT. As a final remark, in the presence of irregular shapes and profiles, as often occurs in LARC, the adaptive ZoT ROIs, markedly different from the tRIM ones, yielded by far the best performance.

However, the study is not devoid of limitations. First, the proposed method for ZoT detection was conceived and implemented in 2D, with the aim of improving the state of art, whose current solutions work in 2D. However, the 3D extension of the method will be pursued in future works, which may even include the development of a fully 3D method for ZoT detection. Second, only the SVM was n used for building the ML models, although it represents a well-established solution to achieving robust results when working with a relatively limited sample size.

## 6. Conclusions

The automated analysis of local image properties allows the ZoT method, in contrast to that of tRIM, not to rely on a prefixed size for peritumour extent, thus enabling peritumour detection even when the region is not visually detectable and manual segmentation not applicable. In addition, defining a peritumour region adaptively favours its clinical interpretability. It should be stressed that the method is compliant with different tumours and imaging modalities, thus contributing to the standardization of the procedures of ML applications in medical image analysis.

From a methodological point of view, future works will explore both a simplification of the entire procedure and alternative implementations of single steps. Finally, from a clinical point of view, we are planning to perform, in the near future, automatic ZoT detection on more anatomic districts, which may even include exploiting public datasets; moreover, we will compare, in a prospective study design, the clinical and biological features of the ZoT regions within surgical specimens.

## Figures and Tables

**Figure 1 sensors-24-01156-f001:**
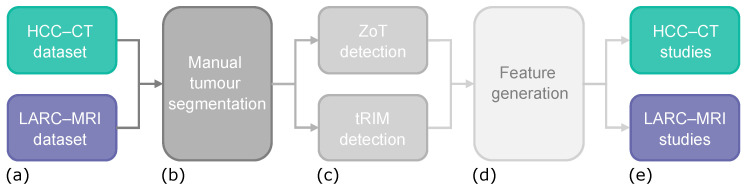
Block diagram showing the main steps of the study’s workflow: input data (**a**) undergo manual tumour segmentation (**b**), that is the starting point for both ZoT and tRIM detection (**c**); then, features are generated from both ZoT and tRIM regions (**d**) and used for predictive purposes in HCC-CT and LARC-MRI studies (**e**).

**Figure 2 sensors-24-01156-f002:**
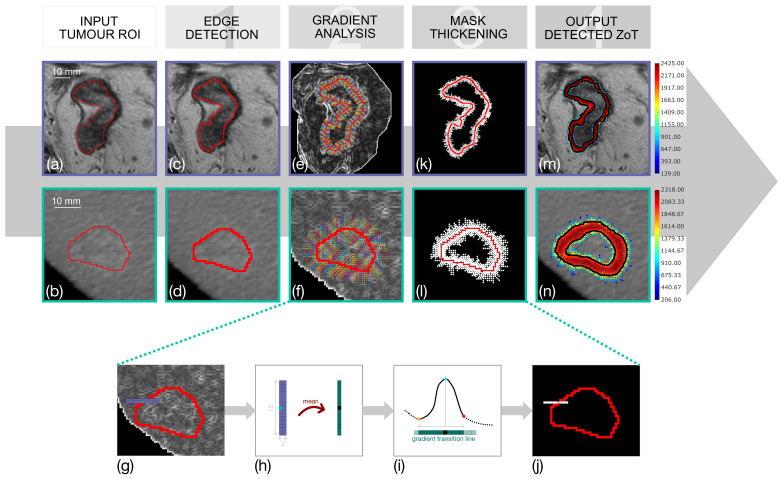
Four-step procedure developed for ZoT detection. The process begins with the segmented tumour ROIs, which stand for the input, shown in (**a**), for LARC and (**b**) HCC. The first step is the morphological edge detection (**c**,**d**), which is followed by the gradient profile analysis (**e**,**f**). In this way, more details are depicted in (**g**–**j**) for a representative edge pixel, analysed along its gradient direction with an assumed rectangular kernel, for instance of 15 × 2 pixels in size. Then, this steps yield binary masks of pixel-based gradient transition zones, which are thickened in the fourth step (**k**,**l**). Finally, the last step is the ZoT detection, arising from the computation of the density maps on the previous masks and their automatic segmentation, which ultimately provides the ZoT’s outer and inner borders (**m**,**n**).

**Figure 3 sensors-24-01156-f003:**
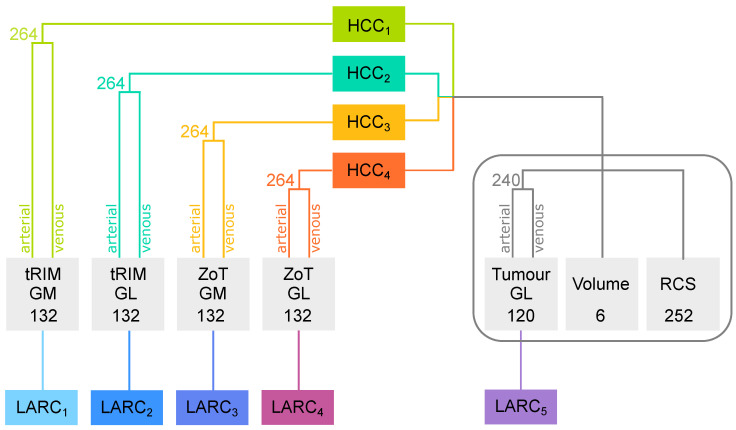
Summary of features considered for the ${\mathrm{HCC}}_{\left[1–4\right]}$ and ${\mathrm{LARC}}_{\left[1–5\right]}$ studies. Each grey box indicates the ROI considered for feature generation (tRIM, ZoT, or tumour) and image type; that is, the gradient magnitude (GM) or grey level (GL). Two more grey boxes refer to volume features and RCS measurements. Each coloured path highlights the pool of imaging features analysed by each study.

**Figure 4 sensors-24-01156-f004:**
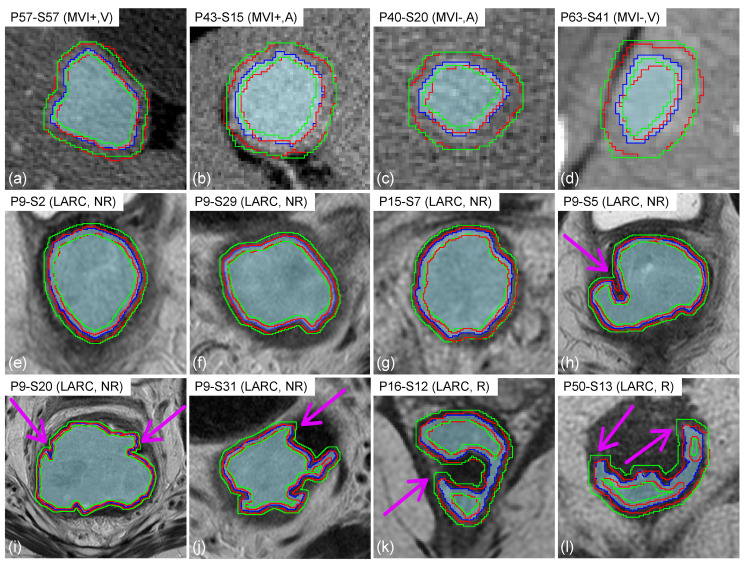
Examples of the ZoT (in red), tRIM (in green), and tumour (the shaded area bounded by the blue line) ROIs. For each image, the patient identification code ($P_{x}$) and slice number ($S_{y}$) are indicated, along with the dataset the samples belong to (i.e., HCC or LARC) and their membership class (MVI+ or MVI− for the HCC dataset; R or NR for the LARC dataset). In addition, the pink arrows highlight the key differences between the tRIM and ZoT ROIs.

**Figure 5 sensors-24-01156-f005:**
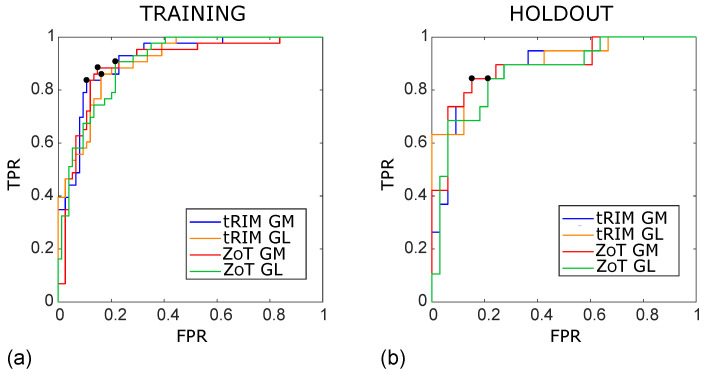
ROC curves of the SVM classifiers arising from the studies ${\mathrm{HCC}}_{\left[1–4\right]}$ achieved for the training (**a**) and holdout (**b**) sets, referring to the tRIM or ZoT analysis performed on the gradient magnitude (GM) or original grey level (GL) images. The outcomes for the training subset were as follows: tRIM GM: AUC ROC = 0.91 (95% CI 0.85–0.96), tRIM GL: AUC ROC = 0.91 (95% CI 0.84–0.95), ZoT GM: AUC ROC = 0.90 (95% CI 0.84–0.94), and ZoT GL: AUC ROC = 0.90 (95% CI 0.83–0.94). The outcomes for the holdout were as follows: tRIM GM: AUC ROC = 0.89 (95% CI 0.77–0.96), tRIM GL: AUC ROC = 0.87 (95% CI 0.74–0.96), ZoT GM: AUC ROC = 0.89 (95% CI 0.75–0.95), and ZoT GL: AUC ROC = 0.87 (95% CI 0.75–0.95).

**Figure 6 sensors-24-01156-f006:**
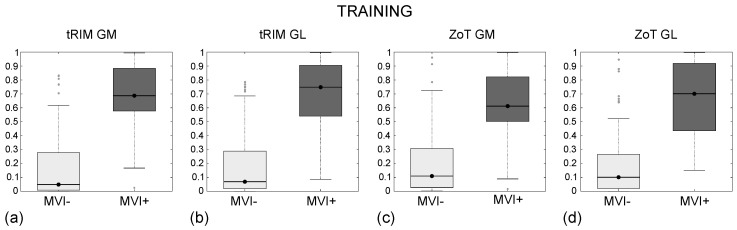
Box plot of the classification achieved by the SVM models in the training sets for the tRIM (**a**,**b**) and ZoT (**c**,**d**) analyses on gradient magnitude (GM) or rgw original grey-level (GL) series. The *p*-values for the Wilcoxon rank-sum test are *p*∼$10^{-14}$ (**a**), *p*∼$10^{-13}$ (**b**,**d**), and *p*∼$10^{-12}$ (**c**).

**Figure 7 sensors-24-01156-f007:**
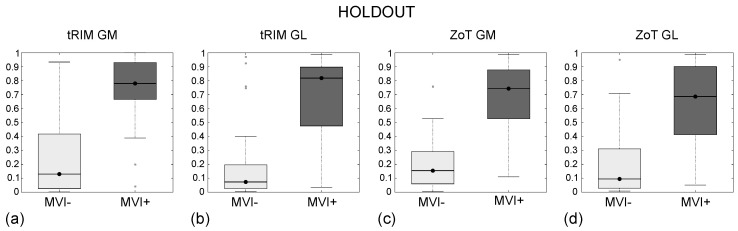
Box plot of the classification achieved by SVM models in the holdout sets for tRIM (**a**,**b**) and ZoT (**c**,**d**) analyses on gradient magnitude (GM) or original grey-level (GL) series. The *p*-values of Wilcoxon rank-sum test are *p*∼$10^{-6}$ (**a**,**c**) and *p*∼$10^{-5}$ (**b**,**d**).

**Figure 8 sensors-24-01156-f008:**
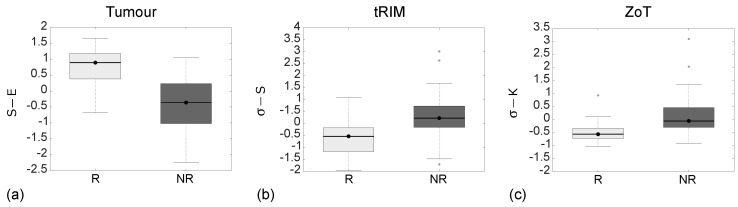
Box plot of S−E (**a**), σ−S (**b**), and σ−K (**c**), referring the t tumour, tRIM, and ZoT analyses for the GM images (${\mathrm{LARC}}_{\left[1,3,5\right]}$ studies) for discriminating the R and NR groups.

**Figure 9 sensors-24-01156-f009:**
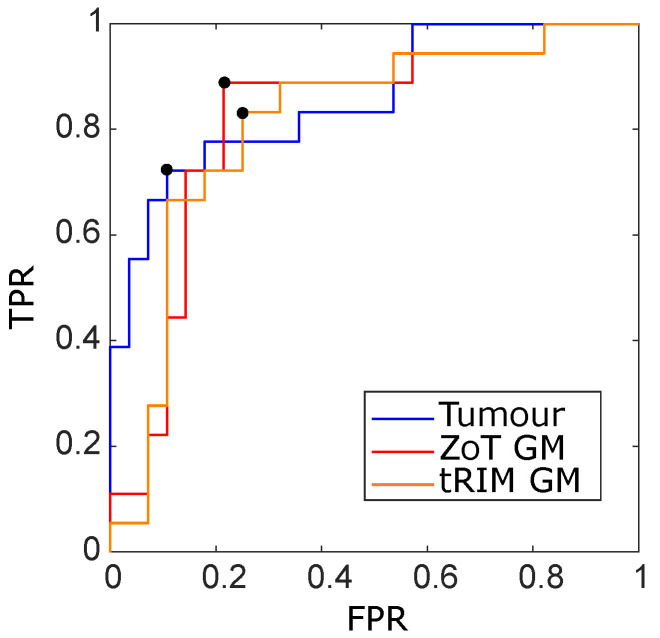
ROC curves of the most discriminant features between the R and NR groups arising from the analysis of tumour (blue line), tRIM (orange line), and ZoT (red line) on gradient magnitude (GM) images, with the Youden cutoff which defines the I value highlighted by black circles. The tumour’s ROC yielded an AUC = 0.86 (95% CI, 0.63–0.92) and an I = 0.62, tRIM’s ROC an AUC = 0.81 (95% CI, 0.65–0.91) and an I = 0.58, and ZoT’s ROC an AUC = 0.82 (95% CI, 0.63–0.93) and an I = 0.68.

**Table 1 sensors-24-01156-t001:** HCC dataset description.

Study Population	Number of Samples
True-positive class	MVI+
True-negative class	MVI−
Initial dataset	89 (32 MVI+, 57 MVI−)
Oversampled dataset (OD)	169 (62 MVI+, 107 MVI−)
OD training set	117 (43 MVI+, 74 MVI−)
OD test set	52 (19 MVI+, 33 MVI−)

**Table 2 sensors-24-01156-t002:** LARC dataset description.

Study Population	Number of Samples
True-positive class	Responders (R)
True-negative class	Non responder (NR)
Study population	46 (18 R, 28 NR)

**Table 3 sensors-24-01156-t003:** Summary of the feature selection performed on all the studies ${\mathrm{HCC}}_{\left[1–4\right]}$. Columns report from left to right the study’s identifier, the number of features initially selected by LASSO (${\mathrm{FS}}_{1}$), all quadruples (${\mathrm{FS}}_{2}$), the uncorrelated quadruples (${\mathrm{FS}}_{3}$), the significant quadruples in the Wilcoxon rank-sum test with Holm-Bonferroni correction (${\mathrm{FS}}_{4}$), the finally selected quadruples (through the single feature identifiers), and their corresponding I.

	${\mathrm{FS}}_{1}$	${\mathrm{FS}}_{2}$	${\mathrm{FS}}_{3}$	${\mathrm{FS}}_{4}$	Quadruple	I
${\mathrm{HCC}}_{1}$: tRIM GM	19	3876	1951	26	[F136, F256, F592, F754]	0.67
${\mathrm{HCC}}_{2}$: tRIM GL	18	3060	1893	21	[F136, F466, F592, F754]	0.67
${\mathrm{HCC}}_{3}$: ZoT GM	16	1820	767	18	[F72, F136, F465, F754]	0.74
${\mathrm{HCC}}_{4}$: ZoT GL	13	715	268	40	[F136, F465, F688, F754]	0.70

**Table 4 sensors-24-01156-t004:** Association between the identifier (ID) and the feature name. In particular, A or V indicate whether the feature refers to the arterial or venous phases, respectively. If the abbreviation “T” is reported as the feature that is computed on tumour ROIs; otherwise it originates from tRIM or ZoT ROIs depending on the study (${\mathrm{HCC}}_{\left[1–4\right]}$) from which it has been selected.

ID	Feature Name
F72	S - E (A)
F136	E - K (T, A)
F256	E - K (V)
F465	K - MAD (T, V)
F466	K - IQR (T, V)
F592	RCS of K − IQR
F688	RCS of μ−IQR (T)
F754	RCS of MAD − K (T)

**Table 5 sensors-24-01156-t005:** ROC-related metrics achieved by the SVM classifiers in the training sets for the prediction of MVI+.

	AUC	I	SN	SP	TP	TN	FP	FN	ACC	NPV	PPV	DOR
HCC_1_	0.91	0.73	84%	89%	36	66	8	7	87%	90%	82%	42
HCC_2_	0.91	0.70	86%	84%	37	62	12	6	85%	91%	76%	31
HCC_3_	0.90	0.74	88%	85%	38	63	11	5	86%	93%	78%	43
HCC_4_	0.90	0.69	91%	78%	39	58	16	4	83%	94%	71%	35

**Table 6 sensors-24-01156-t006:** ROC-related metrics achieved by the SVM classifiers in the test sets for the prediction of MVI+.

	AUC	I	SN	SP	TP	TN	FP	FN	ACC	NPV	PPV	DOR
HCC_1_	0.89	0.69	84%	85%	16	28	5	3	88%	90%	76%	30
HCC_2_	0.87	0.69	84%	85%	16	28	5	3	88%	90%	76%	30
HCC_3_	0.89	0.69	84%	85%	16	28	5	3	88%	90%	76%	30
HCC_4_	0.87	0.63	84%	79%	16	26	7	3	81%	90%	70%	18

**Table 7 sensors-24-01156-t007:** ROC-related metrics achieved by S−E, σ−S, and σ−K referring to the tumour, tRIM, and ZoT analyses (${\mathrm{LARC}}_{\left[1,3,5\right]}$ studies).

	AUC	I	SN	SP	TP	TN	FP	FN	ACC
${\mathrm{LARC}}_{1}$	0.86	0.62	89%	72%	13	25	3	5	83%
${\mathrm{LARC}}_{3}$	0.81	0.58	83%	75%	15	21	7	3	78%
${\mathrm{LARC}}_{5}$	0.82	0.67	89%	79%	16	22	6	2	83%

## Data Availability

Data are not available due to considerations of patients’ privacy.

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
