# Peer review of "The Effectiveness of an Adaptive Method to Analyse the Transition between Tumour and Peritumour for Answering Two Clinical Questions in Cancer Imaging"

_sensors, 2024, doi:10.3390/s24041156_

Round 1

Reviewer 1 Report

Comments and Suggestions for Authors

1. The authors are advised to review the last paragraph, where a few sections appear to be missing (Section ?? discusses the experimental results and Section ?? draws conclusions).

2. In the "Materials and Methods" sections, it is recommended to insert a table describing the datasets, including the number of populations, training and testing data, etc.

3. The authors are encouraged to create a comprehensive block diagram that thoroughly illustrates the complete methodology.

4. The term "ML models" has been mentioned; it is suggested to specify all the Machine Learning Models used (for example, SVM, DT, RF, MLP, BPNN, etc.).

5. It is suggested to include a state-of-the-art table recently published and compare the proposed results with them.

6. Additionally, the authors should include the training curves of the proposed ML models to validate the presented outcomes.

Comments on the Quality of English Language

Throughout the manuscript, there are grammatical and typographical errors. The authors are urged to correct these issues.

Author Response

RESPONSES TO THE COMMENTS

Hereafter, “C#” stands for Reviewer’s “Comment” and “R#” are our Replies (in red). Similarly, the changes in the manuscript are reported in red as well.

REVIEWER #1

Specific Comments

[C1.1] The authors are advised to review the last paragraph, where a few sections appear to be missing (Section ?? discusses the experimental results and Section ?? draws conclusions).

[R1.1] Thank you, references in the last paragraph have been corrected.

[C1.2] In the "Materials and Methods" sections, it is recommended to insert a table describing the datasets, including the number of populations, training and testing data, etc.

[R1.2] Thank you for your suggestion. Table 1 describing the HCC dataset has been added in Section 3.5.2. and Table 2 referred to LARC dataset has been added in Section 3.5.3. 

[C1.3] The authors are encouraged to create a comprehensive block diagram that thoroughly illustrates the complete methodology.

[R1.3] Thank you very much for your suggestion. It has been added as Figure 1, within the new Section 3.2.

[C1.4] The term "ML models" has been mentioned; it is suggested to specify all the Machine Learning Models used (for example, SVM, DT, RF, MLP, BPNN, etc.)

[R1.4] Actually, we have employed the term “ML models” in paragraph #3.5 as it is a general explanation of the workflow stage, whilst we specified “SVM classifiers” in the relative subparagraph’s title of Section 3.5.2 where we deepen the topic and provide implementation details. However, following reviewer’s suggestions we have amended “ML models” into “SVM classifiers” throughout the manuscript, where appropriate.

[C1.5] It is suggested to include a state-of-the-art table recently published and compare the proposed results with them.

[R1.5] We have decided not to include a table comparing our results with the state-of-the-art ones since, to the best of our knowledge, all works found in the literature have substantial differences within study populations which prevent a fair numerical comparison of results. Accordingly, we have decided to apply on our own study population the standard approach of the state-of-the-art, that is the morphological enlargement of tumour ROIs to investigate peritumoral tissue, and our proposed approach, to compare fairly the two methodologies on the same dataset.

[C1.6] Additionally, the authors should include the training curves of the proposed ML models to validate the presented outcomes.

[R1.6] Thank you for suggestion. In cases of a limited population size, like that in this study, the training curve may result poorly significant due to the sparsity of the feature space, thus possibly jeopardizing the evaluation of the training process. For this reason, we decided to focus results' validation on ROC-related parameters and confusion matrices of training and test subsets. Nevertheless, we control the generalization of the model also through monitoring training and test performance metrics for each k-fold run, discarding all models prone to overfitting, before any model selection.

Reviewer 2 Report

Comments and Suggestions for Authors

The overall impression of the technical contribution of the current study is reasonable. However, the Authors may consider making necessary amendments to the manuscript for better comprehensibility of the study.

1. The abstract must be re-written, focusing on the technical aspects of the proposed model, the main experimental results, and the metrics used in the evaluation. Briefly discuss how the proposed model is superior.

2.  Additionally, method names should not be capitalized. Moreover, it is not the best practice to employ abbreviations in the abstract, they should be used when the term is introduced for the first time. For example (ZoT), (LARC).

3. ML is abbreviated multiple times in both the Abstract and introduction.

4. Why to have 10 references to endorse a single statement “priori-defined analytical choices”[6–16], one latest reference is sufficient, and remove the rest of the 9 references. Similarly, [17–21].

5. The introduction section must discuss the technical gaps associated with the current problem.

6. The contribution of the current study must be briefly discussed as bullet points in the introduction. Motivation must also be discussed in the manuscript.

7. The section heading state-of-art may be replaced with Literature Review. More relevant studies could be included for example Self-Learning Network-based segmentation for real-time brain MR images through HARIS

8. A thorough proofreading is recommended, authors may check ?? in Section ?? discusses the experimental results, and Section ??

8. what is this CT [6,7,11–13,16,19–21,24,25,27] and MRI [8,14,15,17,18,26,28]? Is it really required to have those many references here?

9. Authors may clarify tRIM (HCC[1,2] ) or ZoT (HCC[3,4] ) why numbers are presented in sub-script.

10. More explanation of the proposed model is desired on technical grounds.

11. Authors may present the loss functions for better comprehensibility of each of the models used in the proposed model.

12. Please discuss more on the implementation platform and the dataset details as two sub-sections in the manuscript.

13. Authors must provide the details of hyperparameters like training loss, testing loss, training accuracy, and testing accuracy.

14. What are the cases assumed as TP, TN, FP, FN (confusion matrix) in the current study. For better ideas refer to studies like XAI framework for cardiovascular disease prediction using classification techniques

15. More comparative analysis with state-of-art models is desired.

16. By considering the current form of the conclusion section, it is hard to understand by MDPI Journal readers. It should be extended with new sentences about the necessity and contributions of the study by considering the authors' opinions about the experimental results derived from some other well-known objective evaluation values if it is possible.

17. If possible please include ablation study/ Diagnostic odds ratio to extend the results.

Comments on the Quality of English Language

English proofreading is strongly recommended for a better understanding of the study. Few sentences are written in passive voice and it is also observed that a few sentences stop abruptly. 

Author Response

RESPONSES TO THE COMMENTS

Hereafter, “C#” stands for Reviewer’s “Comment” and “R#” are our Replies (in red). Similarly, the changes in the manuscript are reported in red as well.

REVIEWER #2

General Comment

[C2.0] The overall impression of the technical contribution of the current study is reasonable. However, the Authors may consider making necessary amendments to the manuscript for better comprehensibility of the study.

[R2.0] We thank the reviewer for her/his appreciation and provide a point-by-point reply below.

Specific Comments

[C2.1] The abstract must be re-written, focusing on the technical aspects of the proposed model, the main experimental results, and the metrics used in the evaluation. Briefly discuss how the proposed model is superior.

[R2.1] Done, a new abstract is provided.

[C2.2] Additionally, method names should not be capitalized. Moreover, it is not the best practice to employ abbreviations in the abstract, they should be used when the term is introduced for the first time. For example (ZoT), (LARC).

[R2.2] We have rewritten the abstract minimizing the use of abbreviations and defining them at their first mention when strictly necessary. We agree that abbreviations can make the reading difficult, although being sometimes helpful, especially when some requirements on maximum words number have to be fulfilled.

[C2.3] ML is abbreviated multiple times in both the Abstract and introduction.

[R2.3] We have limited abbreviations in the abstract, meanwhile keeping them throughout the manuscript to reduce redundancy of common and recurrent expressions. However, ML is removed from abstract.

[C2.4] Why to have 10 references to endorse a single statement “priori-defined analytical choices”[6–16], one latest reference is sufficient, and remove the rest of the 9 references. Similarly, [17–21].

[R2.4] We have amended the manuscript accordingly.

[C2.5] The introduction section must discuss the technical gaps associated with the current problem.

[R2.5] Technical gaps have been just briefly mentioned in Introduction between lines 36-42, because we have introduced Section 2 “Literature review” to explain technically the approaches proposed in the literature, thus widely contextualizing the need for our adaptive method.

[C2.6] The contribution of the current study must be briefly discussed as bullet points in the introduction. Motivation must also be discussed in the manuscript.

[R2.6] We have claimed the contribution of the current study in Introduction between lines 43-53, although choosing not to use bullet points to emphasize the causality relationships between our innovations. The work has also been contextualized at the beginning of Discussions, between lines 379-382 by recalling the technical gaps of literature approaches.

[C2.7] The section heading state-of-art may be replaced with Literature Review. More relevant studies could be included for example Self-Learning Network-based segmentation for real-time brain MR images through HARIS

[R2.7] Thank you for your comment, the title of Section 2 has now been modified. The aim of this Section was to present the standard approach of the literature to analyse specifically the peritumour region, and the Zone of Transition between tumour and peritumour, so references have been limited to fully pertaining scientific papers. 

[C2.8] A thorough proofreading is recommended, authors may check ?? in Section ?? discusses the experimental results, and Section ??

[R2.8] Thank you for suggestion. Grammatical errors have been corrected as well as the missed references in the last paragraph of Introduction.

[C2.9] what is this CT [6,7,11–13,16,19–21,24,25,27] and MRI [8,14,15,17,18,26,28]? Is it really required to have those many references here?

[R2.9] CT and MRI are abbreviations for Computed Tomography and Magnetic Resonance Imaging, defined at their first mention in Introduction. We used so many references because, in that paragraph of Section “Literature Review” we catalogued all contributions by imaging modalities.

[C2.10] Authors may clarify tRIM (HCC[1,2] ) or ZoT (HCC[3,4] ) why numbers are presented in sub-script.

[R2.10] We guess the reviewer is referring to lines 221. If this is the case, numbers are sub-script because they refer to the denomination of the four studies conducted on HCC dataset, as introduced at line 219-220: “As far as the HCC-CT dataset is concerned, we have set up four different studies (hereinafter HCC[1-4]).”

[C2.11] More explanation of the proposed model is desired on technical grounds.

[R2.11] We dedicated Section 3.3 to describe the methodology of the proposed approach, providing in our opinion all technical details needed to reproduce the method in the whole by researchers familiar with computer vision and image processing. Nevertheless, we believe that this review has brought a great improvement to the overall clarity of the manuscript.

[C2.12] Authors may present the loss functions for better comprehensibility of each of the models used in the proposed model.

[R2.12] As reported in the manuscript, the results are quite similar with each other and the loss function would be useless to understand which model performs better.

[C2.13] Please discuss more on the implementation platform and the dataset details as two sub-sections in the manuscript

[R2.13] Thank you for suggestion. We have added Tables 1 and 2 for dataset details, as also required in [C1.2]. As regards the implementation platform, we reported in the manuscript between lines 192-193 that “The whole pipeline has been implemented in Matlab (R2019b v.9.7, The MathWorks, Natick, MA, USA).”

[C2.14] Authors must provide the details of hyperparameters like training loss, testing loss, training accuracy, and testing accuracy.

[R2.14] Training and test accuracy have been added in the experimental results.

[C2.15[ What are the cases assumed as TP, TN, FP, FN (confusion matrix) in the current study. For better ideas refer to studies like XAI framework for cardiovascular disease prediction using classification techniques

[R2.15] The number of TP, TN, FP, and FN are already presented in Tables 5 and 6 for training and test, respectively, as regards HCC studies. We have also added the values for LARC studies in Table 7.

[C2.16] More comparative analysis with state-of-art models is desired.

[R2.16] We have decided not to include a table comparing our results with state-of-the-art ones since, to the best of our knowledge, all works found in the literature have substantial differences within study populations which prevent a fair numerical comparison of results. Accordingly, we have decided to apply on our own study population the standard approach of the state-of-the-art, that is the morphological enlargement of tumour ROIs to investigate peritumoral tissue, and our proposed approach, to compare fairly the two methodologies on the same dataset.

[C2.17] By considering the current form of the conclusion section, it is hard to understand by MDPI Journal readers. It should be extended with new sentences about the necessity and contributions of the study by considering the authors' opinions about the experimental results derived from some other well-known objective evaluation values if it is possible.

[R2.17] Thank you for your comment, we have now added Section “Conclusion”

[C2.18] If possible please include ablation study/ Diagnostic odds ratio to extend the results.

[R2.18] The Diagnostic Odds Ratio has been added to the experimental results.

Reviewer 3 Report

Comments and Suggestions for Authors

Dear Authors,

I appreciate your thorough review of the manuscript. I have provided some suggestions to enhance the clarity of the study:

Kindly include a schematic diagram in the introduction section to aid comprehension.

Add a new section, Section 5: "Conclusion."

Please review and correct any grammatical errors and ensure accuracy in the references.

We are particularly interested in your insights on the following questions:

1. How does the presented method contribute to the field of quantitative cancer imaging and early diagnosis?

2. What is the Zone of Transition (ZoT), and how does the novel method detect and investigate this area between tumour and peritumour?

3. In what ways does the proposed adaptive method differ from the common approach of extending tumour segmentation of a pre-defined fixed size, particularly in analyzing gradient variations along tumour borders?

4. What are the applications of the presented method, and how does it perform in predicting microvascular invasion in hepatocellular carcinoma and detecting therapy responding patients in locally advanced rectal cancer (LARC)?

5. What advantages does the adaptive ZoT detection method offer in comparison to the common approach, especially in terms of its adaptability to different tumour shapes and imaging modalities?

Thank you for your attention and cooperation.

Best Regards

Author Response

RESPONSES TO THE COMMENTS

Hereafter, “C#” stands for Reviewer’s “Comment” and “R#” are our Replies (in red). Similarly, the changes in the manuscript are reported in red as well.

REVIEWER #3

General Comment

[C3.0] Dear Authors, I appreciate your thorough review of the manuscript. I have provided some suggestions to enhance the clarity of the study:

[R3.0] We thank the reviewer for her/his warm appreciation.

Specific Comments

[C3.1] Kindly include a schematic diagram in the introduction section to aid comprehension.

[R3.1] Thank you for suggestion, we have now added a comprehensive block diagram as Figure 1.

[C3.2] Add a new section, Section 5: "Conclusion."

[R3.2] Thank you for your comment, now we have Section 6 “Conclusion”.

[C3.3] Please review and correct any grammatical errors and ensure accuracy in the references.

[R3.3] Thank you, we have carefully reread the manuscript.

[C3.4] We are particularly interested in your insights on the following questions:

[R3.4] Thank you for your interest. We have provided point-by-point replies from R3.4.a. to R3.4.e.

  1. How does the presented method contribute to the field of quantitative cancer imaging and early diagnosis

[R3.4.a.]  As reported in “Conclusion” (Section 6), our method enables “the peritumour detection even when the region is not visually detectable, and manual segmentation not applicable”, thus also favouring “its clinical interpretability”. Not least, our findings show that, in many cases, the features generated from our Zone of Transition retain a higher information content than that derived from peritumour analysed through the standard approach, thus finally contributing to improve early diagnosis, with a higher predictive performance.

  1. What is the Zone of Transition (ZoT), and how does the novel  method detect and investigate this area between tumour and peritumour?

[R3.4.b.]  This topic has been deepened in Section 3.3, where the novel method for ZoT detection has been presented in detail. To summarize, the ZoT has been defined as “[…] an annular-like shaped area surrounding tumour, where tissue characteristics keep the highest uncertainty”. Our study presents a novel method able to automatically detect the ZoT, adaptively, based on single-patient tumour properties analysed slice-by-slice in CT or MRI image series. To this aim, we studied and detected the highest image contrast variations along tumour borders.

  1. In what ways does the proposed adaptive method differ from the common approach of extending tumour segmentation of a pre-defined fixed size, particularly in analysing gradient variations along tumour borders?

[R3.4.c.]  As shown in current Figure 4 and discussed in Section 4, “[…] in the presence of” tumour ROIs showing “small concavities”, the standard approach (i.e., tRIM) “[…] cannot follow the anatomical profile” and, even worse, “it may include adjacent structures of a different nature that could yield misleading features”.

  1. What are the applications of the presented method, and how does it perform in predicting microvascular invasion in hepatocellular carcinoma and detecting therapy responding patients in locally advanced rectal cancer (LARC)?

[R3.4.d.]  As presented throughout the manuscript, we have applied our novel approach to the prediction of microvascular invasion in HCC and the detection of patients responding to neo-adjuvant chemo-radiation in LARC. In particular, Tables 5 and 6 present the results of MVI prediction in the training and test subsets, where HCC[3-4] studies refer to the ZoT performance on grey-level and gradient magnitude images, respectively. Meanwhile, as regards LARC application, Section 4.3 shows the predictive performance as well as Figure 9 reporting the ROC curves. 

  1. What advantages does the adaptive ZoT detection method offer in comparison to the common approach, especially in terms of its adaptability to different tumour shapes and imaging modalities?

[R3.4.e.]  As also introduced in [R3.4.c], our method for ZoT detection results suitable for all tumour shapes, also including those presenting several “small concavities”, since it follows adaptively the tumour borders by analysing image gradient variations along borders, thus minimizing the risk of including adjacent structures of a different nature, that is very high with the standard approach. In addition, it has been successfully applied on different imaging modalities, thus proving that the gradient study along tumour borders is robust to variations between imaging modalities. Ultimately, we preserve the same applicability to different imaging modalities as the standard approach, but achieving much more specificity and precision in the detected regions.

Round 2

Reviewer 2 Report

Comments and Suggestions for Authors

The authors have fail to address the recommendations of the reviewers in an appropriate manner.

  • The suggestion to avoid abbreviations in the abstract was not heeded, and they continue to be present.
  • The recommendation to present contributions as bullet points for improved comprehensibility has not been implemented.
  • Despite advising for a single latest reference for keywords, the manuscript still displays multiple references for CT and MRI keywords.
  • There is no indication of where the authors have addressed the recommendation to present loss functions for better comprehensibility of each model used.
  • Details about the implementation environment, despite being recommended, are missing.

Author Response

Hereafter, “C#” stands for Reviewer’s “Comment” and R# are our Replies (in red). Similarly, the changes in the manuscript are reported in red as well.

REVIEWER #2

General Comment

[C2.0] The authors have fail to address the recommendations of the reviewers in an appropriate manner.

[R2.0] Some recommendation are a matter of style, and up to the authors. Nevertheless, we removed all the acronym by the abstract, although in our opinion this worses its readability. Nonetheless, we did not use bullet points to highlight our contribution.

Specific Comments

[C2.1] The suggestion to avoid abbreviations in the abstract was not heeded, and they continue to be present.

[R2.1] Done, all abbreviations have been now removed.

[C2.2] The recommendation to present contributions as bullet points for improved comprehensibility has not been implemented.

[R2.2] In our previous reply [ROUND1-R2.6] of revision, we stated that we have chosen “not to use bullet points to emphasize the causality relationships between our innovations”. We appreciate your suggestion, this is a matter of style, but this is choice is up to the Authors, for this reason we did not acknowledge your recommendation.

[C2.3] Despite advising for a single latest reference for keywords, the manuscript still displays multiple references for CT and MRI keywords.

[R2.3] Done.

[C2.4] There is no indication of where the authors have addressed the recommendation to present loss functions for better comprehensibility of each model used.

[R2.4] In the previous reply (round 1) we stated that:

[ROUND1-R2.12] As reported in the manuscript, the results are quite similar with each other and the loss function would be useless to understand which model performs better.

In general, in cases of a limited population size, like that in our study, the evaluation of training loss function may result poorly significant due to the sparsity of the feature space, thus possibly jeopardizing the evaluation of the training process. For this reason, we decided not to discuss the loss functions and rather to focus results' validation on ROC-related parameters and confusion matrices of training and test subsets. Nevertheless, we control the generalization of the model also through monitoring training and test performance metrics for each k-fold run, discarding all models prone to overfitting, before any model selection.

[C2.5] Details about the implementation environment, despite being recommended, are missing.

[R2.5] The implementation environment is Matlab, for which we have provided details between lines 194 and [230-231], now providing also release and version:

  • Line 194: The whole pipeline has been implemented in the Matlab environment (R2019b v.9.7, The MathWorks, Natick, MA, USA)
  • Lines 230-231: As well as for all image processing procedures, the implementation environment for ML models and statistical analyses was Matlab.